# ‘ZOOMing’ in on Consulting with Children and Parents Remotely to Co-Create Health Information Resources

**DOI:** 10.3390/children10030539

**Published:** 2023-03-11

**Authors:** Holly Saron, James Munro, Rob Young, Enitan D. Carrol, David Porter, Ruth Cantwell, Claire Crouch, Julia Roberts, Bernie Carter

**Affiliations:** 1Faculty of Health, Social Care and Medicine, Edge Hill University, Ormskirk L39 4QP, UK; saronh@edgehill.ac.uk; 2Mister Munro Ltd., Liverpool L17 0BT, UK; james@mistermunro.co.uk; 3Faculty Associate at NHS Research & Development North West, Salford HG2 0HD, UK; rob@robyoung.art; 4Alder Hey NHS Foundation Trust, Liverpool L14 5AB, UK; edcarrol@liverpool.ac.uk (E.D.C.); david.porter@alderhey.nhs.uk (D.P.);; 5Institute of Infection, Veterinary and Ecological Sciences, University of Liverpool, Liverpool L69 3BX, UK

**Keywords:** children, co-creation, consultation, digital methods, health information, outpatient parenteral antimicrobial therapy (OPAT), qualitative methods

## Abstract

The COVID-19 pandemic altered the way many people worked. Remote and creative ways were favoured and utilised for consultation activities. In this paper, we draw attention to how we have used creative methods over the teleconferencing platform ‘ZOOM’ to consult with children and their parents when we were unable to consult with them face-to-face. We document a clear timeline of how we have worked together to co-create an animation and information sheet about receiving outpatient parenteral antimicrobial therapy (OPAT). We identify the opportunities and challenges we faced.

## 1. Introduction

In the UK, as elsewhere, during the COVID-19 pandemic, many people were directed to stay home [1] and work from home if they could [2]. Most academic researchers were required to work from home. The disruption of the pandemic meant qualitative researchers, especially those relying on face-to-face interaction with participants, faced new challenges [3,4]; some were forced to re-design or halt their fieldwork. Many researchers within the National Health Service (NHS) in the UK were forced to pause or completely suspend their studies due to restrictions imposed at the local and national level [5]; this was mirrored globally.

Remote, safe, ethical and innovative ways of working needed to be adopted for work to progress [6]. The required shift from face-to-face consultation to remote working created challenges, especially as creativity and flexibility were required [7,8,9]. Institutions and organisations had to reconsider their position on remote ways of conducting research, typically resulting in reduced resistance to and opening up of opportunities to undertake remote research. Paradoxically, as qualitative researchers were ‘locked down’ by the constraints of the pandemic, it generated exciting opportunities, including the utilisation of advances in secure communication technology [10,11]. This allowed them to draw on their own abilities to develop creative, remote research spaces [12] that, pre-lockdown, were likely to be suppressed in favour of more analytical and ‘scientific’ methods [13]. Lockdown also meant that the general public, including children, who are key to qualitative consultation and research, had become more adept at communicating remotely to stay in touch with friends and family and engaging with telecommunication platforms for education, schoolwork [14] and research [15]. It was against this background of a more liberal consideration of remote engagement with children that the shift from face-to-face ‘real-world’ working to remote working became more feasible.

Consultation is part of the wider endeavour of fostering public and patient involvement and engagement (PPIE) [16,17] and integral to many studies focusing on issues of concern to children and young people’s health services [18,19], including the design of information materials [20]. Children are keen to be involved in consultation and research [21]. Consultation within child health is a process of bringing together children, parents and health professionals to enhance health care and to explore experiences and exchange knowledge [22]. Consultation is not research and is not subject to ethical approval or research governance, although it does need to be undertaken ethically. Providing it is not tokenistic, consultation can be extremely valuable as it can ensure that research and resources are underpinned and informed by the experiences of those who will benefit most [18,23,24,25]. One such area relates to children’s experiences of the outpatient parenteral antimicrobial therapy (OPAT) service. OPAT is a service delivered either in a hospital clinic or in the child’s home by community nurses who administer intravenous (administered directly into the bloodstream via a vein) antibiotic medication to treat a potentially serious bacterial infection [26]. Children using the OPAT service are selected by the medical team, who assess their condition and medical history to ensure they are stable enough to be cared for at home.

The remote consultation activity described in this paper came about following a recommendation in a previous research project that identified a need for child-friendly information about OPAT [26]. The consultation activity had been planned to use face-to-face, arts-based, co-creation workshops with children and their parents at a children’s hospital. However, lockdown meant the consultation team had to re-design and adapt their plans to meaningfully engage remotely with children and their parents. Consideration of how children and their parents could utilise creative methods to co-create resources using a telecommunication platform was necessary.

The consultation team consisted of two female academic researchers (one a children’s nurse, the other a social scientist), a professional illustrator and a professional writer (both male).

In this paper, we describe how children and parents were recruited, how we remotely conducted our creative consultation activities using Zoom (a video connection platform) and how we have, collectively, co-created and developed health information resources. We present some of the strengths and challenges of working in this way.

## 2. Materials and Methods

### 2.1. Aim

The aim of the remote consultation activity was to inform and co-create child-friendly health information resources (animation and information sheets) for children receiving OPAT, and parents.

Ethics approval was not required as this work was defined as a Patient and Public Involvement and Engagement (PPIE) consultation exercise. The research study that underpinned the need for this consultation and co-creation of resources was previously approved (NW Greater Manchester West Research Ethics Committee 16/NW/0440).

### 2.2. Recruitment

The clinical team at the study hospital (a tertiary children’s hospital in the UK) identified children who had recently received OPAT at home. They made initial contact with the children’s parents and, if parents agreed, their contact details were shared with the consultation team. A total of 11 parents, with children aged from <1 year old to 17 years old, agreed to be contacted. After receiving parents’ contact details, a member of the consultation team (HS) contacted them, via text message, introducing the work and asking if they would be happy to receive an invitation to the consultation activity via email. Seven parents who were happy for the clinical team to share their contact details either did not respond to the first invitation or further contact from the consultation team or stated that they did not wish to participate (lack of availability, upcoming holidays or other commitments).

Following initial interest in the text message, a specially designed invitation was sent via email to the four parents who sustained interest, and their children. The invitation used images produced by the illustrator to familiarise children and their parents with the style of drawings that would be produced during the online activity (Figure 1). The inclusion of the drawings sparked some interest, and two of the parents asked to see more of the illustrator’s work. Links to previous animations and other health-related information leaflets were sent as examples via email and received positive feedback.

The invitations provided information about the remote consultation activity, and stated when it would take place, who it would be with and why the consultation activity was being conducted. The invitations included information about the animation and information sheet outputs the team wished to co-create together with them (parents and children). After receiving the invitation, parents contacted the consultation team if they, and their child, were still happy to be involved. They were then sent specially designed activity sheets.

### 2.3. Activity Sheets for Children and Their Parents

Paper-based activity sheets were sent to children and parents before the remote consultation to help them prepare, think about and be creative in the way they presented the information they may want to share with the consultation team, in the informal setting of their home. We also hoped that receiving the activity sheets, in advance of the consultation activity, would help children and parents feel comfortable sharing their ideas and experiences with the consultation team (three adults who were unknown to them). Two activity sheets were produced: one sheet especially designed for children (Figure 2) and the other for parents (Figure 3).

The activity sheets were A4 in size. The children’s activity sheets contained five boxes for children to record their answers, as well as more of the illustrator’s drawings. Three boxes asked children to draw pictures and two boxes asked children for written responses. The parents’ activity sheets had three sections with two questions mirroring those the children were asked.

The drawing tasks were designed to encourage children to create some drawings ahead of the online meeting as the team was concerned that children might find it difficult to draw on a screen without time to think about what they would like to draw. It also meant the team had creative responses from the children that could be utilised during the meeting and in developing the animation or illustrations for the information sheets. This was helpful since we did not know which device (e.g., a phone, a tablet, or a computer) children would be using to connect to the activity or how stable the connection would be during the activity. The option of drawing on paper reduced any potentially important experiences not being shared.

Once the children and parents had completed the activities, they emailed the sheets back to a member of the consultation team (HS). How children chose to engage with the worksheets differed. Some answers provided were drawn, some included monochrome drawings, others were coloured in drawings and some drawings included handwritten annotations from the child or parents. Parents used different approaches to engaging with the task. One parent typed their answers into the sheet (Figure 4), whereas the other parents decided to handwrite their answers (Figure 5). None of the parents drew and the detail in the answers provided also varied. After receiving the activity sheets from children and parents, the consultation team met to discuss the answers and plan the session based on the shared information, thus creating an online consultation activity based on individual documented accounts and personal experiences rather than using a set or pre-empted schedule.

### 2.4. Conducting the Online Activity Consultation

During the consultation, the child’s drawings from their activity sheet were utilised by the illustrator. The illustrator was able to crop the images the child shared and draw around them on the screen, bringing them to life for the consultation exercise. For example, one child chose to draw themselves receiving OPAT via a pump, and this became the character used throughout their online workshop, with the illustrator adding further annotations and illustrations based on what the child said during the consultation (Figure 6).

The consultation team followed the children’s lead and asked them about their drawings and asked follow-up questions to their answers. The format of the activity was informal, conversational, engaging and child-centred and provided children with an opportunity to share their own thoughts and experiences. Two members of the team (HS and BC) conversed with children and their parents during the consultation whilst the illustrator (JM) listened, shared their screen and drew images that brought to life, in real time, the experiences children and parents talked about.

Initially, we had planned for the children to ‘annotate’ the illustrator’s drawings on the screen, as this function is available in Zoom. However, the children found it more intriguing to direct the illustrator to do the drawing whilst using the camera to ‘show and tell’ things of importance to them. For example, one child picked up their iPad and showed the team the room and the table where they sat to have their medicine. ‘Show and tell’ meant the illustrator could amend his drawings to truly reflect the information children were sharing. On one occasion, a child spoke about wanting to return home from hospital to see their pet dog. The illustrator asked the child to draw their dog on paper; the child duly shared their drawing and picked up their dog and showed the team on camera. Their drawing of their dog became part of the illustrator’s developing image. Highly iterative opportunities like this, during the consultation, ensured children’s accounts, drawings and experiences shaped and directed the session and supported co-creation with the team. Although parents provided no drawings, they often commented on the illustrations that were created based on the information they had shared. For example, when they spoke about the amount of equipment they had to bring home from the hospital, the illustrator produced a drawing of a car piled high with boxes with equipment written on the boxes (Figure 7) and the parent said, ‘that is exactly what it was like’.

The illustrator, and other team members, paused regularly and checked whether the drawings accurately depicted the experiences that had been shared, asking if any changes needed to be made or if things needed adding. Before the consultation activity ended, the team checked with parents and children that the images were a genuine reflection of what they had shared. This was an important as we wanted to ensure the children and parents were happy with the information as they knew it was the basis for information resources.

During the consultation, the illustrator built an annotated and illustrated ‘map’ of the children and parents’ experiences of returning home and receiving OPAT, and included key words, phrases, people, pets and events that were important to them. This added depth, meaning and further information to the illustrations and helped expand the conversation about their experience of OPAT. A copy of the ‘map’ created was given to the child and their parent as a memento of their contribution (Figure 8). See Appendix A for version of Figure 6, Figure 7 and Figure 8 with handwritten text changed to typed text for ease of reading.

## 3. Results

From our initial contact with 11 parents, some did not respond (*n* = 5), directly declined (*n* = 1) or declined after they had asked questions about the consultation and the activities involved (*n* = 1). A total of four parents (all mothers) with children aged <1 year old, 7 years old, 9 years old and 15 years old accepted the invitation and agreed to participate. Two boys and two girls took part in the consultation, although one boy was only <1 year old; therefore, three children actively engaged in the consultation. The children were still receiving or had received OPAT for different infections and for different durations of time; for example, one child only received one dose at home, whereas another was receiving multiple doses. Different experiences and exposure to OPAT will have influenced the perceptions of the children and their parents. Table 1 provides an overview of the key information learned from undertaking the consultation.

### 3.1. Gaining an Understanding about being on OPAT at Home

Children and their parents shared an abundance of valuable information both in the activity sheets and during the online consultation. They gave key insights into what it is like to receive OPAT at home and the information they received and thought other children and parents should receive. Their experiences were mostly positive and highlighted the strong relationships they had with their community nurses. From the four online consultations via Zoom, the children and parents made it clear that the core information that needed to be included was who was in charge of their child’s care when they left the hospital, who administered the medication and where in the home this would happen, the timing of the medication and why this might be different from the hospital routine. They also thought it was important to know how to look after a child’s line, protect it and keep it safe and who to contact when and if there were any issues.

### 3.2. Creating Impactful Resources from Virtual Consultations with Children and Their Parents

Following the online consultation session, the team liaised closely with children and their parents about the development of the animation and information sheets. We were mindful that not every child or parent would wish to be part of each aspect of the consultation activity, and we were respectful of their choice in how much or how little they were involved following the activity on Zoom. The draft materials were developed directly from the words used by the children and parents and their illustrations and were emailed to parents to discuss with their child and provide feedback on. For example, children and parents were asked to comment on the ‘style’ of the illustrations. The illustrator opted to reflect the online activity and the drawings some children produced as answers on the activity sheets by using crayon-style drawings that would be used in the animation and on the information sheets. Children and parents were asked to comment on the text that was drafted for the information sheets. Parents and children were often pleased with the work and looked forward to receiving more updates. Changes were minimal but were important; for example, one parent sent pictures of the ‘pump’ that had been spoken about and drawn during the consultation activity, to ensure the final drawing would be more accurate. We created a three-minute-long animation and an information sheet.

To ensure clinical accuracy, further consultation occurred with the specialist team at the children’s hospital. Their feedback was invaluable, and they ensured key messages were accurate, issues not highlighted by the children or parents (e.g., the removal of the ‘line’ from the child’s arm) were included and terminology and information aligned with what professionals would provide alongside the resources. The team changed the word ‘cannula’ to the more frequently used word ‘line’ in the script being developed for the animation. They made changes to how the professionals look; the style of uniform they wear; and the amount of equipment they carry with them on home visits. The specialist team was able to provide further generic information to add depth and accuracy to the materials, such as whether the pump would make noises when administering medicine, so this could be incorporated into the animation.

### 3.3. The Strengths of Conducting Online Workshops with Children and Parents

Useful consultations can take place using the internet [22]. A key strength of remote consultation was that the team was able to conduct valuable work during the COVID-19 pandemic whilst keeping everyone (children, parents and professionals) safe. Conducting online activities instead of face-to-face activities also ensures no cost to participants and no additional time was required for travel to and from the activity. Our choice of platform, Zoom, worked well as parents and children were familiar with it from their prior engagement with family and friends.

The consultation was creative and co-constructed with children who were able to engage and interact on screen in a way that was fun and unusual for them. Involving a non-clinical illustrator brought a new perspective to the work, creating opportunities for us to hear, visualise and consider different interpretations. Children and parents were able to take the lead in what they shared and how this was documented. Seeing the drawings and annotations produced in real time meant they could interact with the team to confirm, correct and build on them to ensure they were a true reflection of the account they wanted to provide. The methods used helped to ensure that children and parents were in control of their level of involvement both in the Zoom activity and subsequent co-development work.

### 3.4. The Limitations of Conducting Online Workshops with Children and Parents

Working in this way was not without its limitations; the team relied on the hospital staff during the pandemic to provide them with contact details for parents. When compared to face-to-face consultations, there were fewer opportunities for children to physically draw, although they were provided with activity sheets prior to the session to help reduce the impact of this. One of the meetings involved more than one family, whereas the other two meetings involved only the mother and child and the team. The small number of mother–child pairs was a limitation, and it would have been beneficial to have had a greater number of children and families take part. Our original intention of having separate two-hour group workshops with children and parents aimed to facilitate ‘real world’ face-to-face interactions and opportunities to sit around a table with drawing and writing materials and physically meet people with similar experiences. We found online consulting with children and parents was a far more intense way of working that required a greater degree of organisation and management and concentration for all involved. After the first session, we reflected and decided to run shorter bespoke sessions for each family (child and mother). This was a better use of the participants’ time but increased the time burden for the team. An important consideration, especially in terms of health and socio-economic inequalities, is that the sessions could only be conducted with those who had access to technology and to adequate Wi-Fi.

## 4. Discussion

Although our work was defined as a consultation activity, the principles used and the activities engaged with resonate with creative qualitative research methods.

### 4.1. Consulting with Children ‘in Their Homes’ Can Be Different to Consulting with Children Who Are ‘at Home’

The literature has previously discussed conducting research with children ‘in their homes’ [27,28,29,30], with ‘home’ being the place where children normally live, with people they call their family [31]. Their own home is their preferred place of care during OPAT [26]: a private rather than a public space where outsiders have to ‘cross the threshold’ to enter [32]. Although home is not a homogenous setting, typically it is seen as a safe and secure space for children away from some of the tensions that could occur in other settings [27]. We had to negotiate to cross a digital threshold rather than a physical one. We argue that although the children in our consultation exercise were ‘*at* home’ during the interactive workshop, the space we created was arguably different from being ‘*in* a child’s home’. We argue this is a strength of conducting consultation in this way as the ‘online’ gaze can be directed or managed by children and their parents to create a more limited or controlled image of the home. This is somewhat different from conducting consultation work ‘*in* a child’s home’, which may be more immersive, provide wider access to the constructed family and individual identities in the home but is inevitably more intrusive.

### 4.2. Children as Valuable Co-Creators of Health-Related Information

The findings of this consultation exercise describe an opportunity to work in a different way with children and their parents to gather their thoughts and experiences to develop resources that can be used and benefit others in similar situations. The consultation exercise, and the thoughts and experiences children and their parents shared, helped to build the co-produced health information resources that were identified as being needed within the original study [26]. Often, researchers describe the methods they use, yet do not document or provide detail on how the methods were applied. There is research to show how researchers have engaged with children and parents [33], developed a study with them [34] or co-constructed health-related information [35]. However, although the field is moving forward [16], there is still a lack of knowledge around what works well when co-producing resources, including health information resources. More work is needed in this area to ensure that involvement of the ‘public’, including children and parents in research and consultation activities, becomes a ‘core value’ of child-related work [36,37].

### 4.3. Online Methods Can Be Used to Consult with Children about Their Health Experiences

There is a growing literature that critically considers online videoconferencing methods for consultation activities or research [38,39,40,41,42], with some considering specific platforms such as Skype [40,43,44,45] or Zoom [38,39,46]. Less literature exists on engaging children in videoconferencing, although platforms, such as Skype [45] and Zoom [47], are being used to generate qualitative interview and other creative data effectively and safely. Work conducted prior to the pandemic has demonstrated that participants find Zoom relatively easy to use, cost-effective and generally rated Zoom above alternative methods such as face-to-face, telephone or other video communication platforms [43]. The pandemic brought with it changes in many aspects of online engagement, including security, ease of use and familiarity [11]. We would encourage other academics conducting consultation activities to document their experiences and explore the utility and functionality of online platforms, such as Zoom, with children to support future consultation and research work and advance innovations in online methods with children. This paper addresses deficits in methodological knowledge and provides a clear timeline to support other qualitative academics facing (and fearing) the ‘messy unknown’ and ‘the first step’ [48] of consulting creatively and virtually, especially with children.

The value of the materials is evidenced by the fact that they are being used within practice to provide children (aged about 4 years and older) and their parents to access to tailored information. The materials provide additional user-friendly information to support informed decision making prior to the parents consenting to OPAT. The findings from the consultation have also informed the nurses in their clinical dialogue with children and parents to ensure they address aspects of OPAT (e.g., the shift to once-a-day medication) that may not have previously been covered.

## 5. Conclusions

The COVID-19 pandemic has had a profound impact on the way consultation is conducted. Lockdown forced this change. It resulted in turning to remote methods to conduct activities originally planned to be conducted face-to-face. However, rather than being limiting, it fostered and inspired new ways of working. Children and their parents were able to engage with and co-create resources with the team. They did this from the comfort of their own homes. Despite the pandemic being the trigger for using remote methods, they were an effective method of engagement and a method we will continue to use in the future, although we acknowledge that some consultations may be better served by face-to-face or a mix of online and face-to-face methods.

Using creative methods and the online platform, we were able to see, experience and understand the spaces of children’s home where they receive OPAT, in ways they could chose, control and direct. They were able to construct and control our view of their experience.

## Figures and Tables

**Figure 1 children-10-00539-f001:**
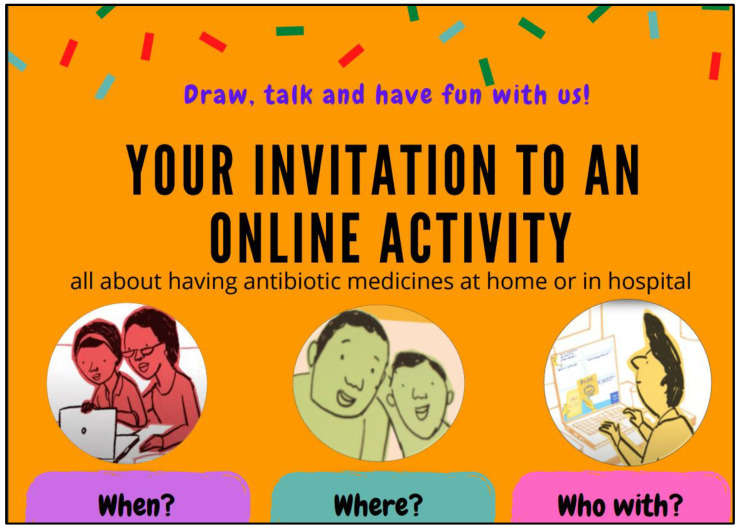
Examples of illustrations on the remote consultation invitation.

**Figure 2 children-10-00539-f002:**
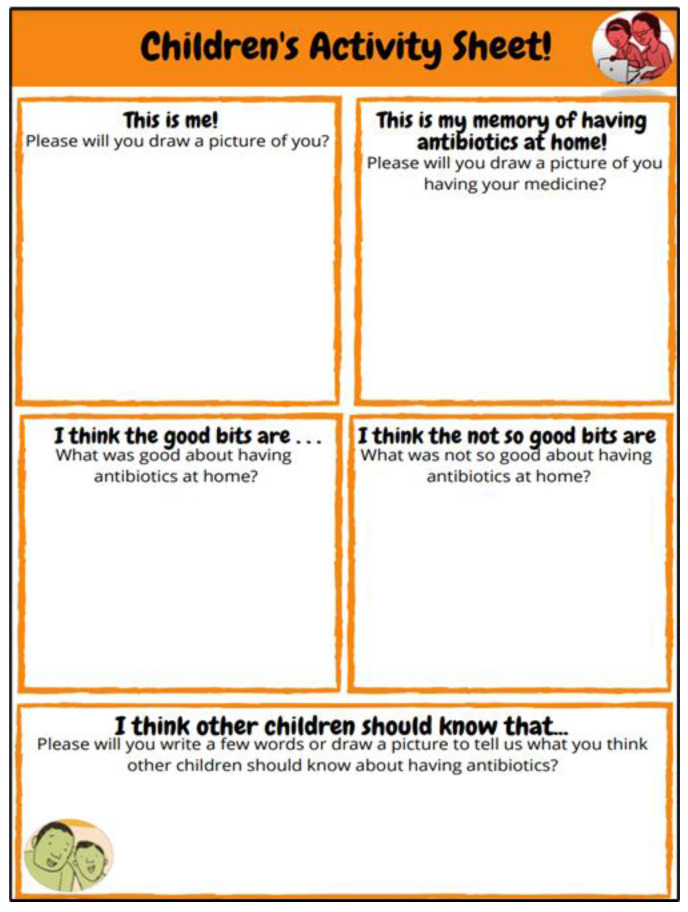
Children’s Activity Sheet.

**Figure 3 children-10-00539-f003:**
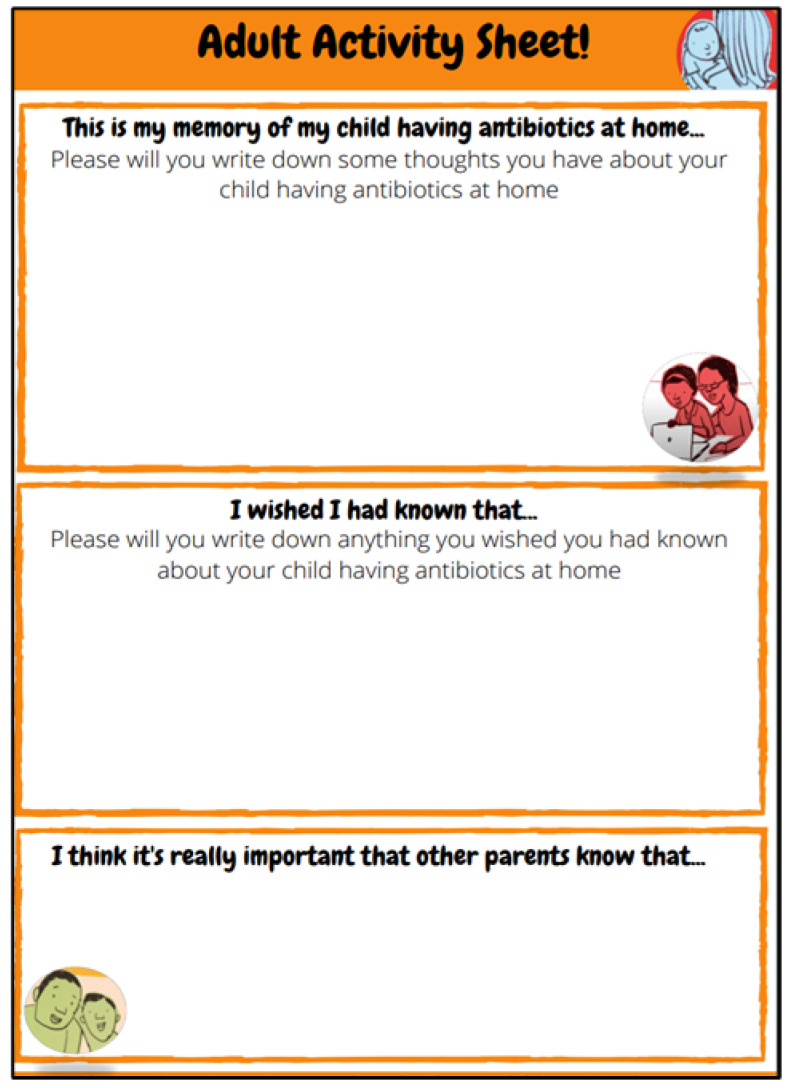
Adult Activity Sheet.

**Figure 4 children-10-00539-f004:**
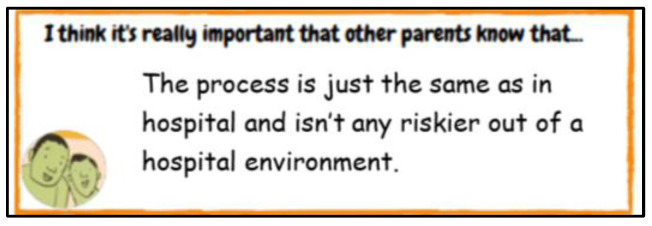
An example of a ‘typed’ response.

**Figure 5 children-10-00539-f005:**
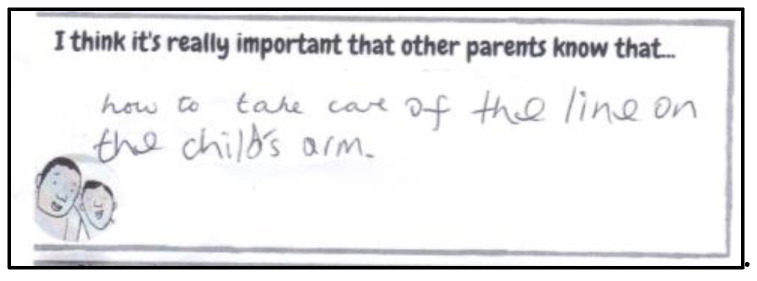
An example of a ‘handwritten’ response.

**Figure 6 children-10-00539-f006:**
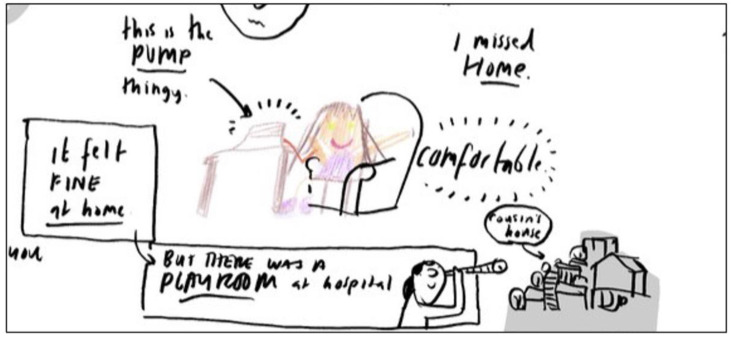
Child’s drawing integrated into the illustration developed during the consultation activity.

**Figure 7 children-10-00539-f007:**
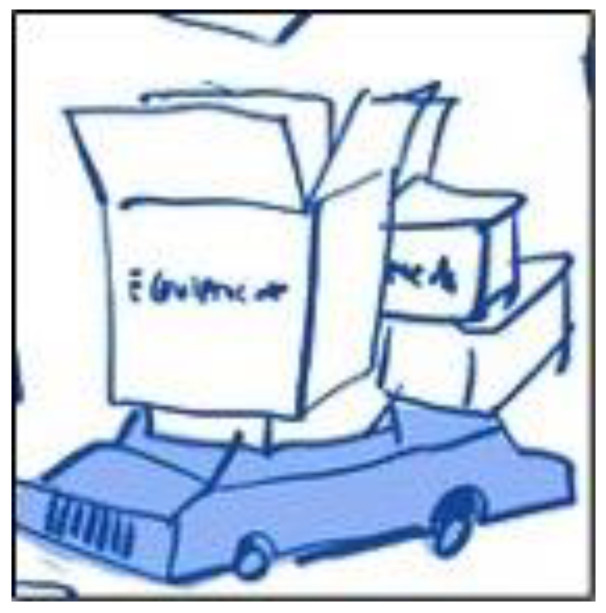
Example of an illustration created from a parent’s account.

**Figure 8 children-10-00539-f008:**
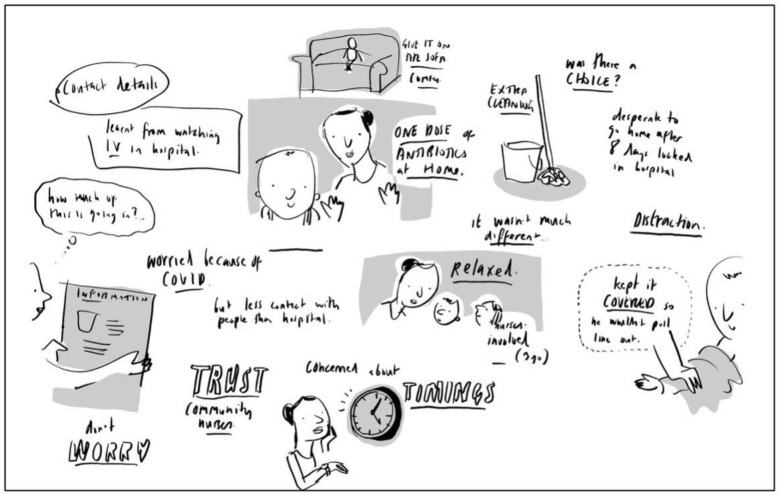
Annotated and illustrated ‘map’ of a child’s experience (child’s name redacted).

**Table 1 children-10-00539-t001:** Overview of key information learned from the consultation activity.

Gaining an understanding about being on OPAT at home	Children’s experiences were mostly positive and were based on having strong relationships with their community nurses. Key information needed included information about how the service was delivered, differences between treatment in hospital and at home, protecting the ‘line’ and who to call for help.
Creating impactful resources from virtual consultations with children and their parents	Key contributions from children and parents included sharing experiences, influencing the design of the illustrations and commenting on drafts of information sheets. Clinical specialists ensured key messages and illustrations were accurate.
The strengths of conducting online workshops with children and parents	Working remotely was low-cost and effective and provided a creative and engaging way to consult with children and parents. Children and parents were in control of their involvement.
The limitations of conducting online workshops with children and parents	Compared to face-to-face consultations, there were fewer opportunities for children to physically draw. Online consulting with children and parents is a more intense way of working compared to real-world engagement. Care needs to be taken not to marginalise or exclude stakeholders without access to technology and/or adequate Wi-Fi.

## Data Availability

The data that underpinned this PPIE consultation are not publicly available.

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
