# Peer review of "‘ZOOMing’ in on Consulting with Children and Parents Remotely to Co-Create Health Information Resources"

_children, 2023, doi:10.3390/children10030539_

Round 1

Reviewer 1 Report

In this manuscript, the authors describe how, during the COVID-19 pandemic, they used a teleconferencing platform (i.e., Zoom) to conduct consultations with three children and four mothers to obtain input and feedback on the “at home” services they received (i.e., outpatient parenteral antimicrobial therapy, OPAT). This information was used to develop health information resources, in this case an educational animation for future families needing these services. The authors concluded that the pandemic led to the development of an innovative way of conducting consultations remotely that will be continued after the pandemic.

This is an interesting case study describing how the team was able develop an innovative framework for conducting consultations via teleconferencing. The number of mother-child pairs is quite small, but seemed to be adequate for developing the educational animated material for future families in need of this procedure.

Suggestions:

It would help if the authors included the small number of mother-child pairs as a limitation.

Line 37. Something is missing in the sentence.  Please double check.

Line 54. The font size needs to be consistent in this line.

Line 83.  Please specify what the acronym PPIE stands for.

Page 6, Figure 6. It’s hard to read the text bubble in lower right – does it say “cousin’s house”?

Page 6, Figure 7. It’s hard to read the text on the boxes – not sure if it is intended to be legible.

Page 7, Figure 8. It would help if the handwritten text was more legible.

Line 254. “team were” should read “team was”.

Reviewer 2 Report

 Usually, we discussed the disadvantages of online activities during COVID. In results, the discussion about the strengths of meetings with students and parents is a good initiative.

More clear result presentation can increase the impact of the findings of the study. The adopted methodology is okay as data collection in such studies is a major constraint. If the result depiction will be in graphical or tabular format, then it will get an easy catch and increase the understanding for the readers. 
